# Variator: Accelerating Pre-trained Models with Plug-and-Play Compression Modules

**Chaojun Xiao[1], Yuqi Luo[1], Wenbin Zhang[1], Pengle Zhang[2], Xu Han[1,3]\*, Yankai Lin[4,5]\*,**
**Zhengyan Zhang[1], Ruobing Xie[6], Zhiyuan Liu[1,3], Maosong Sun[1,3], Jie Zhou[6]**

[1]NLP Group, DCST, IAI, BNRIST, Tsinghua University, Beijing
[2]Zhili College, Tsinghua University, Beijing    [3]Quan Cheng Laboratory
[4]Gaoling School of Artificial Intelligence, Renmin University of China, Beijing
[5]Beijing Key Laboratory of Big Data Management and Analysis Methods    [6]Tencent Inc.

`xiaocj20@mails.tsinghua.edu.cn, hanxu2022@tsinghua.edu.cn, mrlyk423@gmail.com`

## Abstract

Pre-trained language models (PLMs) have achieved remarkable results on NLP tasks but at the expense of huge parameter sizes and the consequent computational costs. In this paper, we propose Variator, a parameter-efficient acceleration method that enhances computational efficiency through plug-and-play compression plugins. Compression plugins are designed to reduce the sequence length via compressing multiple hidden vectors into one and trained with original PLMs frozen. Different from traditional model acceleration methods, which compress PLMs to smaller sizes, Variator offers two distinct advantages: (1) In real-world applications, the plug-and-play nature of our compression plugins enables dynamic selection of different compression plugins with varying acceleration ratios based on the current workload. (2) The compression plugin comprises a few compact neural network layers with minimal parameters, significantly saving storage and memory overhead, particularly in scenarios with a growing number of tasks. We validate the effectiveness of Variator on seven datasets. Experimental results show that Variator can save 53% computational costs using only 0.9% additional parameters with a performance drop of less than 2%. Moreover, when the model scales to billions of parameters, Variator matches the strong performance of uncompressed PLMs. Our code and checkpoints can be found in `https://github.com/thunlp/Compression-Plugin`.

## 1 Introduction

Large pre-trained language models (PLMs) have made significant advancements in natural language processing tasks (Han et al., 2021; Brown et al., 2020; Qiu et al., 2020; Bommasani et al., 2021; OpenAI, 2023). It is widely observed that amplifying the model scale correlates positively with

_______________
\*Corresponding authors.

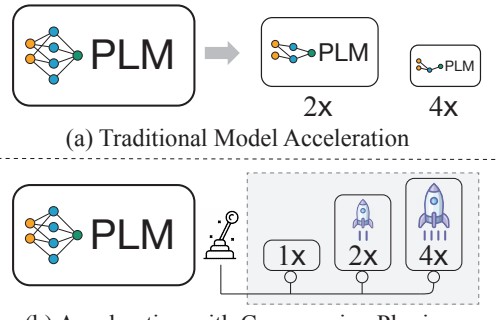

(a) Traditional Model Acceleration

(b) Acceleration with Compression Plugins

Figure 1: Illustration of model acceleration with compression plugins.

enhanced downstream performance. Nevertheless, the expansive parameter scale intrinsic to PLMs demands significant computational and storage resources. Such formidable overheads necessitate investigating alternative strategies to maintain performance while reducing costs.

Many efforts have been devoted to improving the training and inference efficiency of PLMs (Sun et al., 2019; Liu et al., 2022; Fan et al., 2020; Xia et al., 2022; Stock et al., 2021). These methods compress PLMs into fixed smaller sizes, and cannot fulfill the following requirements: (1) Dynamic Workload. In real-world scenarios, the system workload varies dynamically over time, while the computational resources are fixed. This implies that we can use more resources for higher performance when the workload is low, and ensure response efficiency when the workload is high. (2) Storage Efficiency. These methods typically depend on a large number of additional parameters to construct compressed models, which require amounts of memory space for model training and storage across various tasks and acceleration ratios.

To address these issues, we propose a novel plug-and-play acceleration framework named Variator. As shown in Figure 1, different from compressing

PLMs into smaller sizes, Variator enables PLMs acceleration via devising compression plugins, which can be inserted into PLMs to enhance the inference speed. Various plugins entail different acceleration ratios, and the system can dynamically choose the appropriate one to trade off response speed and model performance depending on the workload. Moreover, Variator only necessitates plugins with minimal parameters and freezes the original parameters of PLMs, which substantially lowers the memory and storage requirements.

To achieve plug-and-play acceleration, there are two main challenges: (1) Plugin Architecture: Compression plugins do not modify PLMs scales, and how to devise plugins to reduce the inference time is a challenge. (2) Plugin Training: Compression plugins only contain limited parameters, and how to effectively train plugins so that they can enhance the model speed while preserving downstream performance.

As for plugin architecture, inspired by previous findings about redundancy in hidden vectors (Goyal et al., 2020; Ye et al., 2021), we design compression plugins for data compression rather than parameter compression. Specifically, compression plugins consist of hidden compression layers and hidden decompression layers. The goal of the hidden compression layers is to compress multiple hidden vectors into one, thereby diminishing the sequence length for PLMs and enabling model acceleration. Simultaneously, to preserve token-level information, we also devise decompression layers that recover the processed shorter sequence to the original length. Compression plugins can be applied in any layer of PLMs, enabling various levels of acceleration. As for plugin training, we adopt a two-step training strategy. Firstly, we train compression plugins on pre-trained PLMs with pre-training corpus. Then the compression plugins trained in the first step are used as initialization for task-specific models. In both steps, we apply knowledge distillation objectives to train the compression plugins not to alter the hidden vectors produced by PLMs.

To verify the effectiveness of Variator, we conduct experiments with a widely-used pre-trained backbone, T5 (Raffel et al., 2020), on seven widely-used language understanding benchmarks. The experimental results show that Variator can save 53% computational costs using only 0.9% parameters with absolute average performance drops of $< 2\%$ compared to original downstream PLMs.

When the model scales to billions of parameters, Variator can achieve nearly no performance drop. We also examine the effectiveness of Variator on a decoder-only LLM, LLaMA (Touvron et al., 2023). In addition, we conduct neuron-level analysis for compression plugins, and find that compression plugins can effectively store important information in the compressed vectors to achieve satisfactory performance with limited computational costs.

## 2 Related Work

### 2.1 Model Acceleration

Improving the computational efficiency of PLMs has been widely studied in recent years (Gupta and Agrawal, 2022; Zhang et al., 2022a). The related work can be divided into four categories: knowledge distillation, which guides the training of compressed models with the output or middle states of original PLMs (Hinton et al., 2015; Sanh et al., 2019; Sun et al., 2019, 2020); model pruning, which removes unimportant parameters or layers from PLMs (Fan et al., 2020; Michel et al., 2019; Chen et al., 2020; Xia et al., 2022); model quantization, which converts model parameters into low-bit precision values, thus achieving acceleration on compatible devices (Stock et al., 2021; Xiao et al., 2022); and conditional computation, which only selects parts of parameters to compute outputs for each input (Zhang et al., 2022b; Xin et al., 2020). Some researchers make preliminary exploration for dynamic acceleration, such as early exit (Xin et al., 2020; Matsubara et al., 2023), which attempts to skip layer computation based on instance complexity. But these works rely heavily on confidence judgment and are thus only applicable to specific tasks and model architectures. Our model, which focuses on dynamic acceleration based on system workload, parallels these works and can be intuitively combined with them to reduce computational costs further.

Besides, within the realm of conditional computation there are a line of reasearches find out the redundancy of hidden vectors and focus on discarding tokens at each layer of PLMs to accelerate model inference (Goyal et al., 2020; Ye et al., 2021; Kim and Cho, 2021; Kim et al., 2022; Dai et al., 2020; Murahari et al., 2022), which inspire the design of our compression and decompression layers. But these works require to retrain the whole PLMs to achieve accleration, while Variator focuses on the parameter-efficient acceleration setting and thus

enable dynamic acceleration ratio selection with minimal additional memory requirements. In addition to merge tokens, the compression layer can also be designed to dynamically prune the parameters, which we leave for future work.

## 2.2 Parameter-Efficient Learning

The huge parameter scale of PLMs imposes substantial costs on model training and storage. To alleviate this problem, parameter-efficient learning, also known as delta tuning, is proposed to perform task adaptation via tuning a small portion of parameters and keep other parameters frozen (Liu et al., 2021; Ding et al., 2022; He et al., 2022). According to the operation of tunable parameters, delta tuning methods can be divided into: addition-based models, which introduce additional layers into PLMs (Houlsby et al., 2019; Lester et al., 2021); specification-based models, which specify existing weights of PLMs as tunable (Zaken et al., 2022; Guo et al., 2021); and reparameterization-based models, which rewrite the computation process of specific layers into parameter-efficient manners (Hu et al., 2021). In addition, some researchers attempt to construct plug-and-play modules for retrieval augmentation (Shi et al., 2023; Yu et al., 2023), knowledge injection (Wang et al., 2021; Zhang et al., 2023; Xiao et al., 2023), controllable text generation (Pascual et al., 2021; Madotto et al., 2020), and model debiasing (Lauscher et al., 2021). In this paper, we propose a parameter-efficient acceleration model with hidden vector compression, which can save the memory and storage costs compared to traditional compression methods.

## 3 Methodology

In this section, we first describe the paradigm and basic annotations for our plug-and-play model acceleration. Then we present the framework and training recipes of Variator to accelerate model inference with minimal additional parameters. To showcase the efficiency of Variator, we also conduct an analysis of the computational and storage complexity.

### 3.1 Preliminary

Our primary goal is to design a plug-and-play acceleration framework, which can dynamically improve the computational efficiency with multiple compression plugins. Specifically, given an PLM $\mathcal{M}$, and the fine-tuned downstream model $\mathcal{M}_\mathrm{T}$ derived from $\mathcal{M}$, Variator aims to construct a compression plugin $\mathcal{P}$, which can be inserted into $\mathcal{M}_\mathrm{T}$ to improve the computational efficiency. That is, given an input squence $s$, the computation costs of $(\mathcal{M}_\mathrm{T} + \mathcal{P})(s)$ should be lower than $\mathcal{M}_\mathrm{T}(s)$. Variator is designed for dynamic workload, which means plugins with different acceleration ratios can be applied in the same downstream model $\mathcal{M}_\mathrm{T}$. Therefore, the original $\mathcal{M}_\mathrm{T}$ should be frozen during the training of the compression plugin $\mathcal{P}$.

### 3.2 Overall Framework

Previous researches find out the redundancy in hidden vectors, which means eliminating hidden vectors is a promising direction for acceleration (Goyal et al., 2020; Ye et al., 2021). Inspired by these works, our compression plugins are designed to compress hidden vectors, and thus the sequence length is reduced to speed up inference.

As shown in Figure 2, compression plugins consist of two layers: a hidden compression layer and a hidden decompression layer, which are inserted before and after a vanilla neural layer, respectively. In this way, the compression layer can reduce computational overhead for the following neural layer and the decompression layer aim to restore token-level information into the output vectors. Then we will introduce these two layers in detail.

**Hidden Compression Layer.** Hidden compression layers aim to reduce the sequence length. Previous token pruning methods assign importance scores for each hidden vector and discard hidden vectors with low scores, which may suffer from loss of useful information when the required compression ratio is high. Different from directly dropping hidden vectors, our hidden compression layer is designed to merge multiple vectors into one.

Specifically, given the input vector sequence with $n$ tokens, $\mathbf{H} = \{\mathbf{h}_0, ..., \mathbf{h}_{n-1}\}$, we first split the sequence into several groups with each group containing $k$ vectors, $g_i = \{\mathbf{h}_{ik}, ..., \mathbf{h}_{(i+1)k-1}\}$. Then the compressed vector is calculated as the weighted average of input vectors:

$$\mathbf{a} = \mathrm{Softmax}(\mathbf{W}_c \mathrm{Concat}(g_i) + \mathbf{b}_c),$$

$$\mathbf{g}_i = \sum_{j=0}^{k-1} \mathbf{a}_j \mathbf{h}_{ik+j},$$

where $\mathbf{W}_c \in \mathbb{R}^{k \times kd}(k \ll d)$ and $\mathbf{b}_c \in \mathbb{R}^k$ are trainable parameters, and $d$ is the dimension of hidden vectors. Then the compressed vectors are fed into the original neural layers.

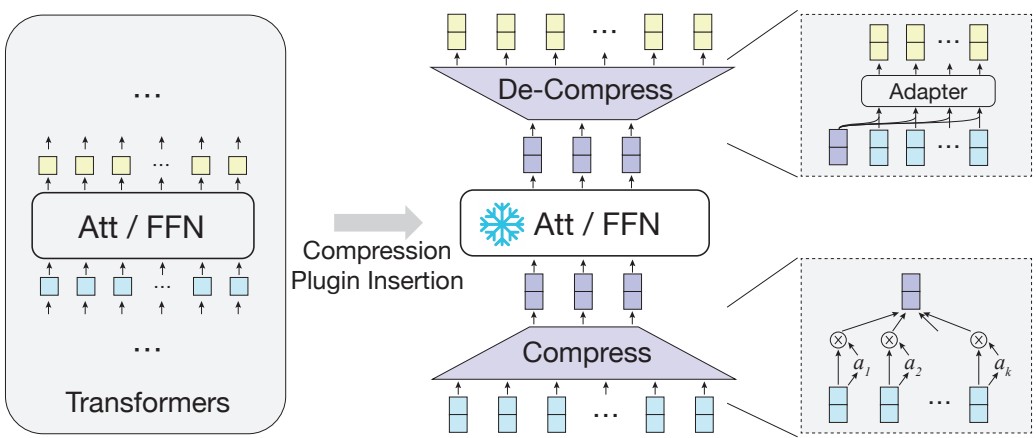

Figure 2: Illustration of Variator, which improves the computational efficiency via compressing the hidden vectors.

**Hidden Decompression Layer.** Compression layers merge multiple hidden vectors into a global vector with information for all tokens in the corresponding group. To preserve the ability to solve token-level tasks, we design hidden decompression layers, which are inserted after the original neural layer, to restore token-level information into the output vectors.

Given the output of the original neural layer $\mathbf{g}_i^o$ generated from the compressed vector $\mathbf{g}_i$, we need to compute the output vectors for all $k$ vectors in $g_i$. We first concatenate the original vector $\mathbf{h}_{ik+j}$ and the compressed output vector $\mathbf{g}_i^o$ to combine the token-level and group-level information. Then, instead of applying a linear projection layer with high computation complexity, we adopt an Adapter (Houlsby et al., 2019) layer and a residual layer to project the concatenated vector to the output vector $\mathbf{o}_{ik+j}$:

$$\mathbf{o}_{ik+j}^{\Delta} = \mathbf{W}_u^2(\mathbf{W}_u^1\text{Concat}(\mathbf{g}_i^o, \mathbf{h}_{ik+j}) + \mathbf{b}_u^1) + \mathbf{b}_u^2,$$
$$\mathbf{o}_{ik+j} = \mathbf{g}_i^o + \mathbf{o}_{ik+j}^{\Delta}.$$

Here, $\mathbf{W}_u^1 \in \mathbb{R}^{r \times 2d}$, $\mathbf{b}_u^1 \in \mathbb{R}^r$, $\mathbf{W}_u^2 \in \mathbb{R}^{d \times r}$, $\mathbf{b}_u^1 \in \mathbb{R}^d$ are trainable parameters, and $r \ll d$ refers to the bottleneck dimension of adapter layers.

Both two layers only involve minimal additional parameters and computation overhead and can significantly reduce the sequence length. Besides, our proposed compression plugins can be flexibly applied in any neural layers, such as self-attention layers and feed-forward layers, allowing for different acceleration ratios. Notably, the compression and decompression layers can be implemented with other efficient operations including convolutional neural networks. Due to the high computa-

tional requirements of feed-forward layers in Transformer (Zhang et al., 2022b), we attempt to apply compression plugins in feed-forward layers in most of our experiments.

### 3.3 Plugin Training

To mitigate information loss during the sequence compression of Variator, we design a two-step training strategy with plugin pre-training and plugin adaptation.

**Plugin Pre-training.** Plugin pre-training aims to learn general information compression ability and obtain a good initialization of compression plugins for downstream models. In this step, compression plugins are trained to mitigate redundancy in the original input text. Specifically, we insert the compression plugins into the original PLM $\mathcal{M}$, and train compression plugins on a pre-training corpus. Notably, the pre-training process is task-agnostic. It is conducted only once and caters to the requirements of all downstream tasks, which make compression plugins pratical even when PLMs scale to billions of parameters.

**Plugin Adaptation.** Plugin adaptation is designed to drive compression plugins to preserve task-specific information during compression. Different tasks tend to pay attention to different information in the sequence. For example, sentiment analysis tasks usually need to maintain the information contained in emotional words, while reading comprehension tasks usually need to maintain information about the question. Therefore, it is important for compression plugins to learn different task information preferences in plugin adaptation. During plugin adaptation, compression plugins are inserted into downstream model $\mathcal{M}_T$, and trained

with task data.

Both steps adopt knowledge distillation as the training objectives, guiding the compression plugins not to modify the output distribution. Given output vectors of the model without compression plugins $\mathbf{O}'$, and output vectors of the model with compression plugins $\mathbf{O}$, the final training loss is computed as the mean squared error (MSE) between $\mathbf{O}'$ and $\mathbf{O}$:

$$\mathcal{L} = ||\mathbf{O}' - \mathbf{O}||_2. \tag{1}$$

### 3.4 Complexity Analysis

In this section, we provide a complexity analysis of computational and storage overhead. Here we present the analysis with compression plugins applied in feed-forward networks (FFNs), with the input length as $n$, hidden vector dimension as $d$, and the middle dimension of FFNs as $4d$. As mentioned in previous sections, $k$ and $r$ refer to the compression ratio and bottleneck dimension of decompression layers.

**Computational Complexity.** Compression and decompression layers involve several linear projections with tiny matrices. Therefore, our compression plugins only require minimal computation costs. For each token, compression plugins contain three linear projection operations and two addition operations. The floating point operations (FLOPs) required by the compression and decompression layer are $(kd+2d+3)n$ and $(3rd+2d+r)n$, respectively. In contrast, the FLOPs of the feed-forward network are $8nd^2$. The computation costs of compression plugins are only about $\frac{1}{8d}(4 + k + 3r)$ of FFN, where $k, r \ll d$. And compression plugins can reduce the computation costs of FFN to $\frac{1}{k}$. Therefore, compression plugins can achieve significant inference speed-up for PLMs.

**Storage Complexity.** Different from training the entire models to accelerate model inference, Variator relies on two projection layers to compress hidden vectors. Compression and decompression layers consist of three linear projection layers, with only $k^2d + k$ and $3rd + r + d$ parameters, respectively. In contrast, an FFN layer consists of $8d^2$ parameters.

To demonstrate the effectiveness of our parameter-efficient compression plugins more intuitively, we assume that $k = 4$, $r = 64$, and $d = 768$. In this way, compression plugins can save $71.7\%$ computational costs with only $3.4\%$ additional parameters for FFNs.

## 4 Experiments

### 4.1 Datasets

To evaluate the effectiveness of Variator, we use seven typical NLP datasets as evaluation benchmarks, including text classification and sequence-to-sequence generation tasks. Specifically, we adopt three natural language inference datasets, MNLI-m (Williams et al., 2018), QNLI (Rajpurkar et al., 2016), RTE (Wang et al., 2019), two sentence similarity datasets, QQP (Wang et al., 2019), MRPC (Dolan and Brockett, 2005), a sentiment analysis dataset, SST-2 (Socher et al., 2013), and a reading comprehension dataset, SQuAD (Rajpurkar et al., 2016). We apply F1 scores for MRPC, F1 scores and exact match scores (EM) for SQuAD, and accuracy for other datasets as evaluation metrics. We also present the average scores on these seven datasets, where we use EM scores of SQuAD for average. Please refer to Appendix for statistics.

### 4.2 Implementation Details

We adopt the widely-used pre-trained model, T5-base and T5-large (Raffel et al., 2020), as our model backbone. Please refer to Appendix for results of compression plugins on BERT backbone (Devlin et al., 2019). For the main experiments, we only insert the compression plugins around the feed-forward network layers in the encoder, which accounts for the majority of computational requirements. As for the training objective, we compute the MSE loss with the output vectors from the last layers. The compression ratio $k$ is set as 4 for the main experiments and the bottleneck dimension $r$ of the Adapter layers is set as 64.

For plugin pre-training, we apply the widely-used Wikipedia corpus. The learning rate is set as $10^{-3}$ and batch size is set as 256. We pre-train compression plugins for 60k steps. For plugin adaptation, we apply grid search for hyper-parameter selection. We select batch size in $\{16, 32\}$, learning rate in $\{10^{-4}, 5 \times 10^{-5}\}$. The total training steps for each task are set as 26k, and we evaluate the models every 1k steps. We train all models with half-precision floating-point on NVIDIA A100 GPUs. For both plugin pre-training and adaptation, we use Adam for parameter optimization. Please refer to Appendix for more details.

### 4.3 Baselines

In this paper, we compare Variator with several competitive baseline models, including: (1) The

| Dataset | MNLI-m Acc. | QNLI Acc. | QQP Acc. | RTE Acc. | SST-2 Acc. | MRPC F1 | SQuAD EM/F1 | Avg. | Para. | FLOPs |
|---|---|---|---|---|---|---|---|---|---|---|
| T5-Base | | | | | | | | | | |
| Original | 86.7 | 93.0 | 91.2 | 82.9 | 94.3 | 92.6 | 82.8/90.0 | 89.1 | – | – |
| Distillation | 84.6 | 91.8 | 89.3 | 81.4 | 93.1 | 93.1 | 81.1/89.3 | 87.8 | 61.9% | 44.3% |
| LTP | 84.0 | 91.7 | 86.5 | 76.8 | 92.6 | 92.5 | 81.1/89.2 | 86.4 | 100.0% | 44.3% |
| Variator | 84.6 | 91.5 | 88.4 | 81.1 | 93.6 | 93.8 | 80.4/88.1 | 87.6 | 0.9% | 46.8% |
| T5-Large | | | | | | | | | | |
| Original | 88.9 | 94.0 | 91.5 | 88.6 | 95.4 | 93.0 | 85.3/92.5 | 91.0 | – | – |
| Distillation | 88.4 | 94.2 | 90.4 | 84.3 | 94.5 | 91.9 | 81.3/90.9 | 89.3 | 59.1% | 52.5% |
| LTP | 87.0 | 93.1 | 88.0 | 82.5 | 94.4 | 93.3 | 84.3/91.7 | 88.9 | 100% | 52.5% |
| Variator | 87.1 | 93.5 | 89.4 | 85.4 | 93.7 | 92.8 | 83.1/90.7 | 89.3 | 0.7% | 54.1% |

Table 1: Comparison results between Variator and baseline models. Here Avg. refers to the average scores on seven datasets. Para. and FLOPs refer to the ratio of the number of additional parameters and floating point operations required by the compressed methods to the original PLMs.

original fine-tuned downstream PLMs without acceleration, which are also used as teacher models to guide the training of other compressed models. (2) The widely used model compression method, model distillation (Sanh et al., 2019). (3) Our method aims to reduce the sequence length for PLMs, which is inspired by previous token pruning models. Therefore, we also compare Variator with a typical token pruning model, LTP (Kim et al., 2022), which adopts the attention scores as importance scores, and only keeps tokens with the most scores for each layer. Notably, original token pruning models directly discard tokens for entire Transformer layers, and our models in main experiments focus on the acceleration of FFNs. Therefore, to make a fair comparison, we implement token pruning models with only skipping computation of FFNs and only keeping $25\%$ tokens in each layer. We apply knowledge distillation objectives to train all downstream tasks for a fair comparison.

## 4.4 Main Results

The comparison results are shown in Table 1. To further demonstrate the effectiveness of Variator, we show the additional parameters, and FLOPs for each input required by compressed models. Here we assume the input length is $512$ and batch size is 1 for calculating FLOPs. From the results, we can observe that: (1) Variator can achieve comparable results with the original PLMs using minimal additional parameters with absolute performance drops of $< 2\%$. Specifically, Variator can save $53.2\%$ and $45.9\%$ computation costs for T5-base and T5-large, using only $0.9\%$ and $0.7\%$ additional parameters. In contrast, traditional acceleration methods need to construct compressed models

| Dataset | MNLI-m Acc. | SST-2 Acc. | SQuAD EM/F1 |
|---|---|---|---|
| Variator | **84.6** | **93.6** | **80.4/88.1** |
| w/o PT | 84.1 | 92.7 | 79.3/87.3 |
| w/o PA | 59.6 | 87.8 | 11.6/19.4 |
| w/o Com | 83.5 | 92.3 | 79.4/87.1 |
| w/o DeCom | 73.8 | 86.1 | 38.2/50.6 |

Table 2: The results for ablation study.

from scratch, which require amounts of additional parameters. Limited by amounts of parameters, switching traditional methods between different compression ratios requires large memory space or repeatedly loading compressed models from the disk. (2) Compared to the widely-used model distillation, our parameter-efficient model acceleration method achieve competitive performance with much fewer parameters, which indicates the potential of parameter-efficient model compression. (3) Compared to the token pruning baselines, our models can achieve better performance with even a small portion of parameters, which proves that merging tokens can better preserve sequence information compared to directly dropping them.

## 4.5 Ablation Study

To verify the effectiveness of each component of Variator, we conduct an ablation study in this section. Specifically, we show the results of compression plugins without plugin pre-training (w/o PT) or plugin adaptation (w/o PA). Besides, we also examine the effectiveness of compression and decompression layers in the ablation study. We show the model performance with compression layers replaced with a mean-pooling operation (w/o Com) or decompression layers replaced with a copy operation (w/o DeCom). We run w/o Com and w/o

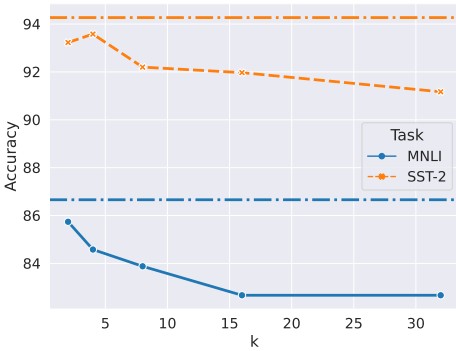

Figure 3: Model performance with different compression ratios. The horizontal lines indicate the performance of original PLMs without compression plugins.

DeCom without plugin pre-training to speed up experiments. We select three tasks for the ablation study, including sentence classification, SST-2, sentence-pair classification, MNLI-m, and reading comprehension, SQuAD.

The results are shown in Table 2. From the results, we can observe that: (1) Both two training steps contribute to the main model, as when anyone step is missing, the model performance drops significantly. (2) Plugin adaptation is important for all tasks. Plugin pre-training guides compression plugins to discard general redundant information contained in the input text. Therefore, for SST-2, which usually only focuses on parts of important words, compression plugins without task-specific adaptation can also achieve satisfactory results. In contrast, for SQuAD and MNLI-m, which require models to collect information from entire contexts, plugins without adaptation lead to a large performance drop. (3) Compression and decompression layers play an important role in selecting information for hidden merging and restoring token-level information, as without anyone of them, model performance drops significantly. Especially, decompression layers are quite important for preserving token-level information, and training compression plugins without decompression layers lead to large drops for the span extraction task, SQuAD.

### 4.6 Effects of Compression Ratios

Variator apply compression plugins to compress multiple hidden vectors into one, thus achieving inference speedup. In this section, we explore the effects of compression ratios for our compression plugins. We construct compression plugins with compression ratios as $\{2, 4, 8, 16, 32\}$. The results are shown in Figure 3.

From the results, we can find that: (1) With

| Dataset | MNLI-m Acc. | SST-2 Acc. |
|---|---|---|
| Original | 86.7 | 94.3 |
| Variator (FFN) | 84.1 | 92.7 |
| Variator (Att) | 81.0 | 91.9 |
| Variator (Att-KV) | 83.1 | 92.1 |

Table 3: The results for compression plugins inserted around the self-attention layers.

the compression ratio increasing, the model performance decreases as expected. But the decline rate is becoming slow, which indicates the potential for Variator to achieve higher compression ratios. (2) Variator can achieve competitive performance even when the compression ratio reaches 32, where Variator maintains $95.4\%$ and $96.7\%$ accuracy scores of original PLMs for MNLI-m and SST-2, respectively, while reducing $69\%$ computational costs. The satisfactory performance ensures the response speed of real-world applications when the system load is high.

### 4.7 Compression for Attention Layers

In our main experiments, we insert the compression plugins around the FFN layers. In this section, we examine the performance of Variator when we insert compression plugins around the self-attention layers. Here we do not perform plugin pre-training. The results are shown in Table 4. For comparison, we also present the results of original models and Variator with plugins in FFN layers.

From the results, we can observe that Variator with plugins in self-attention layers perform worse than plugins in FFN layers. That is because self-attention layers are designed to fuse token-level information, and inserting hidden compression layers before self-attention layers would lead to the loss of token information. Thus in the self-attention layers, only the $k$-gram information integration is performed, resulting in a significant performance drop. To address this issue, we improve compression plugins for self-attention layers by only compressing key and value vectors, denoted as Variator (Att-KV). With compression only for key and value vectors, Variator (Att) can achieve comparable results with Variator (FFN). And compressing key and value vectors can be further adopted in decoder-only models to reduce the sequence length of past key-value vectors, which we leave for future work.

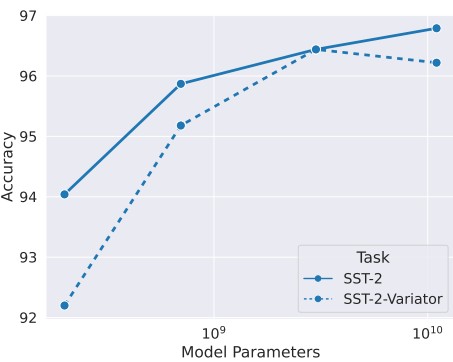

Figure 4: Performance with different backbone sizes.

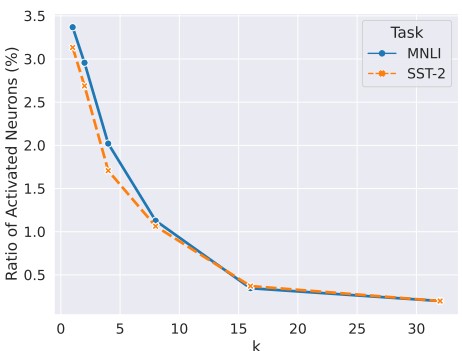

Figure 5: The ratio of activated neurons with different compression ratios on two datasets.

## 4.8 Scaling to PLMs with Billions of Parameters

In this section, we attempt to apply our compression plugins to PLMs with billions of parameters. We adopt four variants of T5 as our backbones, including T5-base (200 million parameters), T5-large (700 million parameters), T5-XLarge (3 billion parameters), and T5-XXLarge (11 billion parameters). Following the main experiments, for each model, we conduct the two-step training process with 6k-step plugin pre-training and 26k-step plugin adaptation. We apply a parameter-efficient learning method, LoRA (Hu et al., 2021), to train the task-specific downstream models to speed up the experiments. We show the results of the SST-2.

As shown in Figure 4, the performance continues to improve with the increasing of backbone model sizes. Similar to previous parameter-efficient learning methods (Lester et al., 2021; Ding et al., 2022), the performance gap between Variator and original PLMs becomes small when the model scales to billions of parameters. It shows the potential of Variator to be applied in nowadays general PLMs with more than 100 billion parameters, such as ChatGPT and GPT-4 (OpenAI, 2023).

| | LLaMA-7B | Variator (w/o PT) |
|---|---|---|
| SST-2 | 97.3 | 96.3 |

Table 4: The results for compression plugins in LLaMA.

Besides, to present the effectiveness of Variator on decoder-only LLMs, we evaluate Variator with recent popular backbone LLaMA (Touvron et al., 2023) with 7 billion parameters. Variator can be used for the input encoding acceleration and reduce the service latency in real-world applications. We conduct experiments with a compression ratio of 2 on the FFN layers and without plugin-pretraining to accelerate experiments. The results suggest that our approach can reduce the computational overhead while maintaining comparable performance with the original model for decoder-only LLMs.

## 4.9 Neuron-Level Analysis

Our compression plugins enable the feed-forward layers to process information from multiple tokens simultaneously to save computational costs. In this section, we attempt to explore the computational mechanism of our compression plugins from the perspective of activated neurons. Previous works find out that FFNs can be regarded as memory networks (Geva et al., 2021; Dai et al., 2022), and the activated neurons can be used as indicators to reflect what information is preserved in the input hidden vectors. T5 adopts ReLU (Nair and Hinton, 2010) as the activation function, and following Zhang et al. (2022b), we define activated neurons as ones with positive (non-zero) activation values.

We present the average ratio of activated neurons with different compression ratios, $k = \{1, 2, 4, 8, 16, 32\}$, in Figure 5. From the results, we can observe that the ratios of activated neurons drop with the increase in compression ratios. When the compression ratio reaches 32, only less than 2‰ neurons are activated to process sequence information. In indicates that compressed hidden vectors only contain the necessary information for the sequences and discard unimportant ones. Besides, the low activated ratios also indicate the potential of the combination of Variator and neuron pruning methods (Zhang et al., 2022b) to further improve the computational efficiency.

Then we further explore the relationship between the activated neurons of FFNs with compression plugins and the activated neurons of original FFNs. Specifically, we denote the intersection set and union set of activated neurons of $k$ hidden vec-

| $k$ | 2 | 4 | 8 | 16 | 32 |
|---|---|---|---|---|---|
| $\|\mathcal{C} \cap \mathcal{I}\|/\|\mathcal{I}\|$ | 0.89 | 0.85 | 0.79 | 0.66 | 0.61 |
| $\|\mathcal{C} \cap \mathcal{U}\|/\|\mathcal{C}\|$ | 0.88 | 0.89 | 0.93 | 0.98 | 0.99 |

Table 5: The relationship between activated neurons of Variator and original models.

tors as $\mathcal{I}$ and $\mathcal{U}$. The set of activated neurons of compressed vector as $\mathcal{C}$. The intersection set $\mathcal{I}$ can be regarded as important global information for $k$ hidden vectors, and $\mathcal{U}$ can be regarded as all information contained in the $k$ hidden vectors. Compression layers are used to select important information and feed them into neural layers. Therefore, we hope that $\mathcal{I}$ is approximately a subset of $\mathcal{C}$ and $\mathcal{C}$ is approximately a subset of $\mathcal{U}$. In Table 5, we present what fraction of neurons in $\mathcal{I}$ are in $\mathcal{C}$ and what fraction of neurons in $\mathcal{C}$ are in $\mathcal{U}$. From the results, we can observe that when the compression ratios are no more than 8, the relationship between the three sets approximately satisfies the abovementioned inclusion assumption. It proves the effectiveness of our compression plugins in preserving global information. When the compression ratios become larger (such as 16, 32), only no more than 70% neurons in $\mathcal{I}$ are contained in $\mathcal{C}$. That is because with the increase of compression ratios, selecting global important information from multiple vectors becomes challenging for compression layers with limited parameters. It also shows the potential to add regularization from the neuron level for compression plugs to preserve important information.

## 5 Conclusion

In this paper, we explore the parameter-efficient acceleration setting and propose Variator, which reduces the computational costs with compression plugins. The extensive experiments on seven datasets show that we can reduce 53% computational costs with only 0.9% additional parameters. In the future, we will explore more effective token-merging frameworks to improve compression plugins. Besides, we will further decouple compression plugins from specific tasks, thus we can construct compression plugins once and for all with transferability across multiple tasks.

## Acknowledgement

This work is supported by the National Key R&D Program of China (No.2022ZD0116312), National Natural Science Foundation of China (No. 62236004), Tsinghua-Toyota Joint Research Fund, and Institute Guo Qiang at Tsinghua University.

**Author Contributions** In the preparation and discussion of the project, Chaojun Xiao, Yuqi Luo, and Xu Han designed the model architectures. Chaojun Xiao and Yuqi Luo wrote the code and conducted the experiments. Besides, Wenbin Zhang and Pengle Zhang wrote the code for baseline models and ablation study. Chaojun Xiao wrote the initial draft. Xu Han, Yankai Lin, Zhengyan Zhang, Ruobing Xie, and Zhiyuan Liu significantly edited and improved the paper. Maosong Sun and Jie Zhou provided valuable advice to the research.

## Limitations

We discuss the limitations of Variator in this section: (1) In the experiments, we implement Variator with T5 as our backbone. It is worth exploring applying Variator in other large-scale decoder-only pre-trained models. (2) In this paper, we mainly focus on accelerating the encoding process of PLMs. Language decoding also plays an essential role in real-world applications. In the experiments, we show the potential of Variator to compress key and value vectors for acceleration. We believe Variator can also serve as a flexible framework to speed up decoding. (3) Our plug-and-play compression framework parallels other model compression methods. It is worth exploring the combination of multiple acceleration methods to achieve more efficient and effective model inference frameworks.

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

## A  Training Details

In this section, we describe some training details, including the datasets and hyper-parameters used in our experiments.

### A.1  Datasets

As for the plugin pre-training corpus, we adopt a widely-used Wikipedia corpus for pre-training. To facilitate the pre-training process, we split each document into several paragraphs with $128$ tokens.

As for the plugin adaptation datasets, we adopt seven widely used language understanding datasets as our evaluation benchmarks. As we use T5 (Raffel et al., 2020) as our backbone, we formalize all these tasks into sequence-to-sequence formats. The detailed statistics and the input template are shown in Table 6.

### A.2  Implementation Details

In this subsection, we describe the implementation details used in our experiments.

As for plugin pre-training, we use $8$ A100 (80G) GPUs to train Variator on T5-base for $4.9$ hours and T5-large for $9.9$ hours. We adopt the knowledge distillation objectives to pre-train plugins. Following settings in Raffel et al. (2020), the mean length of the masked span is set as $3$, and the mask ratio is set as $0.15$.

As for baseline implementation, we fine-tune the original T5 with learning rate searched from $\{10^{-5}, 3 \times 10^{-5}, 5 \times 10^{-5}\}$ and batch size searched from $\{16, 32\}$. The checkpoints with the best validation performance are used as the teacher models to distill all other baselines. For distillation models, we first conduct task-agnostic distillation for $10k$ steps on the Wikipedia corpus, where the learning rate is set as $10^{-4}$ and the batch size is set as $256$. For both distilled models and token pruning models, we fine-tune them on downstream data using distillation objectives, with learning rate searched from $\{10^{-5}, 5 \times 10^{-5}\}$, and batch size searched from $\{16, 32\}$.

## B  Training Objectives

In this paper, we adopt knowledge distillation to guide the training of compression plugins to preserve token-level and sequence-level information. It is intuitive to adopt the task-specific loss function to optimize the parameters of compression plugins. In this section, we explore the effects of task-specific objectives and knowledge-distillation

| Dataset | Train | Validation | Input Template |
|---------|-------|-----------|----------------|
| MNLI-m | 393k | 9.8k | Sentence 1: `PREMISE` Sentence 2: `HYPOTHESIS` Does sentence 1 entails sentence 2? `<extra_id_0>` |
| QNLI | 105k | 5.5k | Question: `QUESTION` Sentence: `SENTENCE` Does the sentence contains the answer to the question? `<extra_id_0>` |
| QQP | 364k | 40.4k | Question 1: `QUESTION1` Question 2: `QUESTION2` Are the two questions paraphrase of each other? `<extra_id_0>` |
| RTE | 2.5k | 277 | Sentence 1: `SENTENCE1` Sentence 2: `SENTENCE2` Does sentence 1 entails sentence 2? `<extra_id_0>` |
| SST-2 | 67.3k | 872 | Sentence: `SENTENCE` Does this sentence express positive or negative emotions? `<extra_id_0>` |
| MRPC | 3.7k | 408 | Sentence 1: `SENTENCE1` Sentence 2: `SENTENCE2` Are the two sentences paraphrase of each other? `<extra_id_0>` |
| SQuAD | 87.k | 10.6k | Question: `QUESTION` Context: `CONTEXT` Answer: `<extra_id_0>` |

Table 6: The statistics and input templates of downstream datasets. In the templates, the task-specific inputs are denoted in `monospaced font`, and `<extra_id_0>` refers to the special mask token for T5.

| Dataset | MNLI-m Acc. | SST-2 Acc. |
|---------|-------------|------------|
| $\lambda = 0$ | 84.6 | 93.6 |
| $\lambda = 0.1$ | 83.3 | 92.2 |
| $\lambda = 0.5$ | 83.1 | 92.4 |

Table 7: The performance with different training objectives.

| Dataset | MNLI-m Acc. | SST-2 Acc. |
|---------|-------------|------------|
| Original | 83.3 | 93.0 |
| Variator (BERT) | 80.0 | 90.3 |

Table 8: The performance with compression plugins in BERT.

## C  Compression Plugins for BERT

Our compression plugins can be applied in Transformer-based pre-trained models. In this section, we explore inserting compression plugins into the widely-used encoder-only pre-trained model, BERT (Devlin et al., 2019). We adopt the 100-million-parameter version, BERT-base, as our backbone. Following the main experiments, we set the compression ratio as 4 and the bottleneck dimension as 64. We conduct plugin pre-training for $24k$ steps. The results are shown in Table 8. From the results, we can observe that Variator on BERT can also show competitive results, and longer plugin pre-training is supposed to lead to better performance.

objectives. Here, we denote the task-specific loss as $\mathcal{L}_t$ and the distillation loss as $\mathcal{L}_d$. The final loss is calculated as $\mathcal{L} = \lambda \mathcal{L}_t + \mathcal{L}_d$. We present the performance of Variator with compression ratio $k$ as 4 on T5-base.

As shown in Table 7, we can find that training with task-specific objectives leads to performance drop on both MNLI-m and SST-2 datasets. That is because task-specific loss is usually easier to optimize than distillation loss, and adding task-specific loss functions makes our compression plugins more likely to fall into the local optimum of the model. Therefore, in other experiments, we only utilize the distillation loss functions to optimize compression plugins.