# OpenReview forum: "Variator: Accelerating Pre-trained Models with Plug-and-Play Compression Modules"
_EMNLP/2023/Conference — EMNLP 2023 Findings_

### Official Review · Reviewer_3yEr · 2023-07-19

**Soundness:** 3
**Typos Grammar Style And Presentation Improvements:** 1. The emergence of "compression plug…

**Excitement:**

2: Mediocre: This paper makes marginal contributions (vs non-contemporaneous work), so I would rather not see it in the conference.

**Paper Topic And Main Contributions:**

Although Large language models have achieved significant performance in many tasks, the enormous computations and memory cost are still a challenging problem. In this paper, the authors propose Variator, a parameter-efficient acceleration method that enhances computational efficiency via a plug-and-play way. Here, they introduce a compression plugin, which reduces the sequence length by compressing multiple hidden vectors into one and then training with frozen LLM.

**Questions For The Authors:**

1. The proposed method seems only suitable for language understanding tasks since it introduces hidden vector compression within a group (i.e., future information). So, how to utilize this method into generation model, especially current LLMs usually refer to generation models, rather than understanding.

**Reasons To Accept:**

The proposed method attempts to accelerate model speed by compressing hidden vectors and thus reducing sequence length.

**Reasons To Reject:**

1. The experiments are only conducted on T5, which lacks enough convincing since the authors demonstrate their experimental settings are for LLMs. Therefore, more LLMs (e.g., LLaMa) should be provided. Besides, it will be better if authors can also provide results on BERT to demonstrate the generalization of the proposed method.
2. The idea of the proposed method (compress sequence) is similar to [a], but this paper does not cite or discuss this paper. Authors should give some discussions about the difference of [a].
3. The proposed method introduces (plugin) pre-training into the training, hence I do not think the proposed method is suitable for large language models (at least 7B like LLaMa) and only suitable for some pre-trained models. Since large language models usually cost enormous memory, it is impractical for LLMs to pre-train. As the target of this paper is for acceleration, using pre-training has violated this policy.
4. Experimental results only compared with distillation and LTP, without any comparison of other compression models (e.g., TinyBERT, MiniLM, etc)

[a] Zihang Dai, Guokun Lai, Yiming Yang, Quoc Le. Funnel-transformer: Filtering out sequential redundancy for efficient language processing

**Reproducibility:**

3: Could reproduce the results with some difficulty. The settings of parameters are underspecified or subjectively determined; the training/evaluation data are not widely available.

**Reviewer Confidence:**

4: Quite sure. I tried to check the important points carefully. It's unlikely, though conceivable, that I missed something that should affect my ratings.

---

> ### Author Rebuttal · Authors · 2023-08-29
>
> Thanks for your insightful suggestion and positive comments.
>
> **R1: More backbones**
>
> In the appendix, we present experimental results on BERT, which similarly confirm that significant inference acceleration can be achieved with a minimal number of additional parameters. In the revised manuscript, we will incorporate experiments using LLaMA as the backbone to further validate the effectiveness of our method.
> Besides, as shown in Section 4.8, we provide experimental results on widely-used LLMs, T5-XLarge (3 billion parameters) and T5-XXLarge (11 billion parameters). The results demonstrate that Variator can achieve promising performance when the model scales to billions of parameters.
>
> **R2: Missing references**
>
> The primary contribution of our paper lies in exploring a novel, plug-and-play approach for model inference acceleration with minimal additional parameters. Sequence compression is one of the mechanisms used to achieve this goal within this specific context. Thus, the distinction between our work and [a] is as follows: [a] focuses on a more efficient pre-training architecture and requires retraining the model from scratch. In contrast, we concentrate on accelerating the inference of existing pre-trained models for downstream tasks, accomplished by employing exceedingly small-scale plugins while preserving the integrity of the original model.
> We will include a discussion on the differences between Viarator and [a] in the revision.
>
> **R3: Plugin pre-training**
>
> As discussed in Line 291-294, the pre-training process in our method is task-agnostic and serves the purpose of finding an appropriate initialization point for the plugin. It is conducted only once and caters to the requirements of all downstream tasks. This process shares similarities with task-agnostic model distillation methods, which is widely employed for LLMs. These methods first obtain a distilled version of a pre-trained model on task-agnostic pre-training data and then adapt it to specific tasks.
> Moreover, the overhead introduced by plugin pre-training is minimal: during this phase, we keep the parameters of the original LLM unchanged and only update the plugin's parameters (which constitute less than 1\% of the LLM's parameters). This reduction in parameter count contributes to a decrease in memory consumption and computational burden. Notably, the plugin pre-training requires only 60k steps, significantly fewer than the nearly one million steps needed for traditional language model pre-training (such as masked language model and autoaggressive language model).
>
> As shown in Table 2, Variator without pre-training can also achieve good performance, which indicates the flexibility and effectiveness of Variator.
>
> In the future, we will release the pre-trained compression plugins to promote future research and application.
>
> **R4: Comparison with other compression models**
>
> The main focus of our paper is on the context of lightweight acceleration achieved through minimal parameter addition. It is foreseeable that traditional compression setups generally involve building a compression model from scratch, demanding substantial additional storage space. This diverges from the specific focus of our work.
> Furthermore, the proposed algorithm can be seamlessly integrated with conventional compression techniques such as distillation, pruning, and quantization, as they serve complementary purposes. In the revised manuscript, we will include additional results that encompass a broader spectrum of traditional compression techniques to offer a comprehensive comparative analysis.
>
> **Q1: Generation tasks**
>
> The algorithm we propose is not limited to language understanding tasks, and it can also be effectively applied to sequence-to-sequence tasks. For instance, in our experiments, we evaluate Variator on question-answering tasks, SQuAD, which we formalize as sequence-to-sequence generation task. This demonstrates the framework's compatibility to reduce the computational overhead of the input encoding process.
> Besides, Table 3 displays the effectiveness of key and value vector compression within attention layers. This approach can also be extended to the decoding process by compressing key and value vectors, thereby reducing the computational burden of attention mechanisms. Consequently, we believe our algorithm possesses a robust versatility that lends itself to various types of tasks, including generation tasks.

---

### Official Review · Reviewer_prtK · 2023-07-25

**Soundness:** 3

**Excitement:**

3: Ambivalent: It has merits (e.g., it reports state-of-the-art results, the idea is nice), but there are key weaknesses (e.g., it describes incremental work), and it can significantly benefit from another round of revision. However, I won't object to accepting it if my co-reviewers champion it.

**Paper Topic And Main Contributions:**

This paper introduces Variator, a parameter-efficient acceleration method designed to enhance the computational efficiency of large language models (LLMs) without sacrificing performance. The proposed approach employs compression plugins that consolidate multiple hidden vectors into a single one, offering dynamic selection of various plugins with different acceleration ratios, depending on the workload. The effectiveness of Variator is validated on seven datasets, demonstrating a 53% reduction in computational costs with only 0.9% additional parameters and less than a 2% performance drop.

**Reasons To Accept:**

The paper is well-written, making it easy to comprehend and follow. The figures' visual quality is exceptional, with the teaser figure being a particular highlight. The proposed method is technically sound and achieves decent empirical performance. The authors have thoughtfully included ample implementation details, enabling other researchers and practitioners in the community to reproduce their work successfully. Furthermore, the authors' commitment to releasing their codebase upon paper acceptance adds to the paper's transparency and facilitates further exploration and validation of their findings.

**Reasons To Reject:**

My major concern with this paper centers around its experimental evaluation, which requires further clarification and elaboration.

* Firstly, upon reviewing Table 1, I observed that the proposed Variator achieves comparable performance to the other two baselines, particularly Distillation, with similar accuracy and #FLOPs. This similarity raises questions about the distinct advantages of Variator over existing methods.
* Secondly, I find it challenging to grasp why the proposed Variator claims to compress the model size. From my understanding, Variator introduces additional transformation layers to the original model, which should theoretically increase the model size instead of reducing it. The paper needs to provide more clarity on this aspect.
* Thirdly, it is important to note that #FLOPs (floating-point operations) represent a hardware-agnostic metric. A reduction in #FLOPs may not necessarily equate to reduced latency on specific hardware. To address this limitation, I recommend the authors report measured latency on hardware in both tables and figures, providing a more comprehensive evaluation of the proposed method's real-world performance.

Additionally, I noticed that the proposed method bears similarities to token merging, where similar tokens are merged to save computation, as effectively demonstrated in the computer vision community. In light of this resemblance, the authors should offer both verbal and quantitative comparisons with token merging to highlight the uniqueness and advantages of Variator.

**Reproducibility:**

4: Could mostly reproduce the results, but there may be some variation because of sample variance or minor variations in their interpretation of the protocol or method.

**Reviewer Confidence:**

3: Pretty sure, but there's a chance I missed something. Although I have a good feel for this area in general, I did not carefully check the paper's details, e.g., the math, experimental design, or novelty.

---

> ### Author Rebuttal · Authors · 2023-08-29
>
> Thanks for your insightful suggestion and positive comments.
>
> **R1: Advantages of Variator**
>
> The primary objective of this paper is to enhance the flexibility of compression algorithms while reducing the additional storage overhead required by model acceleration techniques.
> Specifically, this work focuses on a novel plug-and-play acceleration paradigm, which aims to introduce only a small number of additional parameters to achieve model inference acceleration, enabling dynamic selection of acceleration ratios, and reducing storage overhead.  The experimental results in Table 1 demonstrate that Variator can achieve plug-and-play model acceleration with only 0.9\% additional parameters, and the performance is comparable to traditional acceleration methods which require large additional storage for the compressed models trained from scratch.
>
> **R2: Model size compression**
>
> As disscussed in Section 3.1, Variator is not designed for compressing the model size; rather, it introduces a minimal additional module to achieve inference acceleration. The traditional approaches, such as distillation, necessitate constructing different models for different tasks and acceleration ratios, incurring substantial storage overhead. For instance, considering an LLM with 10 billion parameters, obtaining compressed models for N tasks at computation reduction of 1/2, 1/3, and 1/4 would require creating 3N compressed models, each ranging from 2.5 to 5 billion parameters. In contrast, Variator achieves acceleration with less than 1\% additional parameters for each task and acceleration ratio, significantly reducing storage overhead and the plug-and-play characteristic enables dynamic selection of acceleration ratios.
>
> **R3: Latency reduction**
>
> We further conduct latency testing to assess the reduction brought about by Variator. Our supplementary tests reveal that for T5-large, Variator reduces the FFN computation latency from 0.18s to 0.103s (including compression, decompression, and FFN computation). This demonstrates a tangible enhancement in model computational efficiency. We will incorporate a discussion on latency in the revised version.
> Moreover, as part of our ongoing efforts, we plan to integrate specific efficient computational operators that align with the Variator framework. This step is anticipated to result in even more pronounced improvements in latency reduction.
>
> **R4: Comparison with token merging**
>
> Thank you for your valuable suggestion. As previously mentioned in Q1, Variator aims to enhance the flexibility and storage efficiency of acceleration algorithms. To this end, we utilize a token merging algorithm to implement compression plugins. Token merging techniques commonly applied in computer vision could potentially be adapted for implementing compression and decompression layers, which we leave for future work. In the revision, we will provide a discussion on the differences between Variator and token merging algorithms in CV community.

---

### Official Review · Reviewer_rvsY · 2023-08-11

**Soundness:** 4

**Excitement:**

3: Ambivalent: It has merits (e.g., it reports state-of-the-art results, the idea is nice), but there are key weaknesses (e.g., it describes incremental work), and it can significantly benefit from another round of revision. However, I won't object to accepting it if my co-reviewers champion it.

**Paper Topic And Main Contributions:**

This paper presents a parameter-efficient acceleration approach called Variator, which reduces the computational costs with compression plugins by compressing input tokens/hiddens into a compressed representation to shorten the length. The extensive experiments on seven datasets show that Variator can reduce 53% computational costs with only 0.9% additional parameters.

**Questions For The Authors:**

1. What if you use CNN as the module to shorten the sequence length?

**Reasons To Accept:**

1. The idea is smart and interesting, with very limited additional overhead.
2. The experimental results demonstrate its effectiveness across many datasets.

**Reasons To Reject:**

1. The method reminds me of using a convolutional layer with stride > 1 (e.g., =k) to process the input sequence. The author should compare to this.

**Reproducibility:**

4: Could mostly reproduce the results, but there may be some variation because of sample variance or minor variations in their interpretation of the protocol or method.

**Reviewer Confidence:**

4: Quite sure. I tried to check the important points carefully. It's unlikely, though conceivable, that I missed something that should affect my ratings.

---

> ### Author Rebuttal · Authors · 2023-08-29
>
> Thanks for your insightful suggestion and positive comments.
>
> **R1: Comparison with convolutional layers**
>
> Thank you for your suggestion. We explored the use of a convolutional neural network as a compression layer in preliminary experiments, and the results were comparable to our current compression approach. We will address this and provide additional discussion on this aspect in the revised version.

---

### Meta-Review · Area_Chair_g3wd · 2023-09-18

**Recommendation:** 3

**Metareview:**

This paper introduces Variator, a parameter-efficient acceleration method for large language models (LLMs) that aims to reduce computational costs and enhance efficiency. Variator utilizes compression plugins to compress multiple hidden vectors into a single representation, shortening the sequence length. Experiments conducted on seven datasets show a 53% reduction in computational costs with only a 0.9% increase in parameters and less than a 2% performance drop.

Strengths:
* The idea is novel and interesting, with limited additional overhead.
* The method is technically sound and demonstrates effectiveness across various datasets.
* The paper is well-written and provides sufficient details to reproduce the results.

Weaknesses:
* The method has similarities to existing approaches, such as convolutional layers with stride > 1 and token merging, and requires comparisons with these methods.
* The experimental evaluation needs further clarification and elaboration, including comparisons with other compression models and exploring the method's applicability to a wider range of LLMs.
* The proposed method may not be suitable for large language models requiring pre-training, as it may violate the acceleration objective.
The method's suitability for generation models needs to be addressed.

Overall, the paper has merits and potential and the weaknesses and concerns raised by the reviewers are partially addressed.

---

### Decision · Program_Chairs · 2023-10-07

**Decision:**

Accept-Findings

**Comment:**

This paper introduces Variator, a parameter-efficient acceleration method for large language models (LLMs) that aims to reduce computational costs and enhance efficiency. Variator utilizes compression plugins to compress multiple hidden vectors into a single representation, shortening the sequence length. Experiments conducted on seven datasets show a 53% reduction in computational costs with only a 0.9% increase in parameters and less than a 2% performance drop.

Strengths:
* The idea is novel and interesting, with limited additional overhead.
* The method is technically sound and demonstrates effectiveness across various datasets.
* The paper is well-written and provides sufficient details to reproduce the results.

Weaknesses:
* The method has similarities to existing approaches, such as convolutional layers with stride > 1 and token merging, and requires comparisons with these methods.
* The experimental evaluation needs further clarification and elaboration, including comparisons with other compression models and exploring the method's applicability to a wider range of LLMs.
* The proposed method may not be suitable for large language models requiring pre-training, as it may violate the acceleration objective.
The method's suitability for generation models needs to be addressed.

Overall, the paper has merits and potential and the weaknesses and concerns raised by the reviewers are partially addressed.